



# An online emission module for atmospheric chemistry transport models: Implementation in COSMO-GHG v5.6a and COSMO-ART v5.1-3.1

Michael Jähn[1], Gerrit Kuhlmann[1], Qing Mu[1,a], Jean-Matthieu Haussaire[1], David Ochsner[1], Katherine Osterried[2], Valentin Clément[3], and Dominik Brunner[1]

[1]Swiss Federal Laboratories for Materials Science and Technology (Empa), Dübendorf, Switzerland
[2]Center for Climate Systems Modelling (C2SM), ETH Zurich, Zurich, Switzerland
[3]Federal Office for Meteorology and Climatology (MeteoSwiss), Kloten, Switzerland
[a]now at: Norwegian Meteorological Institute, Division for Climate Modelling and Air Pollution, Oslo, Norway

**Correspondence:** Dominik Brunner (dominik.brunner@empa.ch)

**Abstract.** Emission inventories serve as crucial input for atmospheric chemistry transport models. To make them usable for a model simulation, they have to be pre-processed and, traditionally, provided as input files at discrete model time steps. In this paper, we present an "online" approach, which produces a minimal number of input data read in at the beginning of a simulation and which handles essential processing steps online during the simulation. For this purpose, a stand-alone Python package "emiproc" was developed, which projects the inventory data to the model grid and generates temporal and vertical scaling profiles for individual emission categories. The package is also able to produce "offline" emission files if desired. Furthermore, we outline the concept of the online emission module (written in Fortran 90) and demonstrate its implementation in two different atmospheric transport models, COSMO-GHG and COSMO-ART. Simulation results from both modeling systems show the equivalence of the online and offline procedure. While the model run-time is very similar for both approaches, disk storage and pre-processing time are greatly reduced when online emissions are utilized.

## 1 Introduction

Gridded and temporally varying emission fields are a critical input for three-dimensional atmospheric chemistry transport models (Matthias et al., 2018). Traditionally, such emissions are obtained from an inventory and pre-processed for the model as external inputs, which are read in during simulation at regular time steps. The pre-processing typically includes a mapping of the inventory to the model grid and the application of temporal and vertical profiles depending on emission source category. In case of atmospheric chemistry models, a speciation, i.e. a mapping of the species in the inventory to those simulated in the model is required, for example the mapping of total non-methane volatile organic compounds (NMVOCs) to individual NMVOC species. A prominent example of an emission pre-processing model is the Sparse Matrix Operator Kernel Emissions model (SMOKE, https://www.cmascenter.org/smoke/), which has been used extensively with different atmospheric chemistry models in the United States (McHenry et al., 2004; Wong et al., 2012), Europe (Borge et al., 2008; Bieser et al., 2011a),





and Asia (Wang et al., 2011). More recent examples are the pre-processor PREP-CHEM-SRC (Freitas et al., 2011) and the High-Elective Resolution Modelling Emission System HERMES Guevara et al. (2019).

In case of a large number of tracers and hourly input, the pre-processing and reading of the emission fields (hereafter referred to as "offline approach") imposes a large computational burden in terms of input and output (I/O) operations and

storage. An alternative is to apply the temporal and vertical profiles inside the model ("online approach"), which greatly reduces the number and size of input files and the corresponding I/O. This approach enhances the flexibility in setting up new model simulations since no new emission data need to be generated before the start of a simulation. Furthermore, meteorology-dependent emissions, such as emissions from residential heating depending on outdoor temperatures, can be incorporated more easily.

Here we present such an online emission module, which has been integrated into two different atmospheric transport models, both based on the regional numerical weather prediction and climate model COSMO (Consortium for Small-scale Modeling; Baldauf et al., 2011). The first model was developed for the passive transport of tracers such as long-lived greenhouse gases, and is referred to as COSMO-GHG (Liu et al., 2017; Brunner et al., 2019). The second model is COSMO-ART (Aerosols and Reactive Trace gases; Vogel et al., 2009; Knote et al., 2011), which simulates the transport and chemistry of reactive gases

and aerosols as well as two-way feedbacks between meteorology and chemistry. Both are fully integrated, online coupled models simulating the meteorology and the transport (and chemistry) of atmospheric constituents consistently in one single model (Baklanov et al., 2014). The module has also been incorporated into the model COSMO-ART-M7 (Glassmeier et al., 2017), which extends COSMO-ART with an optional simplified chemistry and aerosol scheme for climate applications, but this implementation will not be discussed here.

The purpose of this study is to present the conceptual framework of the online emission module, to demonstrate its suitability and flexibility for the simulation of greenhouse gases and air pollutants, and to show its equivalence to the traditional offline approach. Although the implementation of the module is specific to COSMO, the overall concept is generic and the code has been written in a modular way to facilitate integration into other (Fortran-based) model systems, for example as an extension to the recently developed emission module for ICON-ART (Weimer et al., 2017). The online module is composed of one

Fortran module file, which interfaces with several routines of the standard COSMO model and its extensions GHG and ART as described later.

Furthermore, a separate Python package generates a small number of input files required for the simulation. Different from the offline approach, these files are read in only once at the beginning of a simulation. The Python package also supports the generation of hourly input files for offline applications. This capability is used here to compare simulations with online

and offline emission processing. In contrast to the Fortran module, the Python package is independent of the specific model implementation and, therefore, of potential use for any atmospheric transport model system.

The manuscript is organized as follows: Section 2 describes the functionality of the Python pre-processing package. Section 3 presents the implementation of the online emission module in the modeling systems COSMO-GHG and COSMO-ART and demonstrates its advantages based on three practical examples. Finally, Section 4 shows a performance assessment for both

offline and online processing and demonstrates the equivalence of the two approaches based on two example simulations.





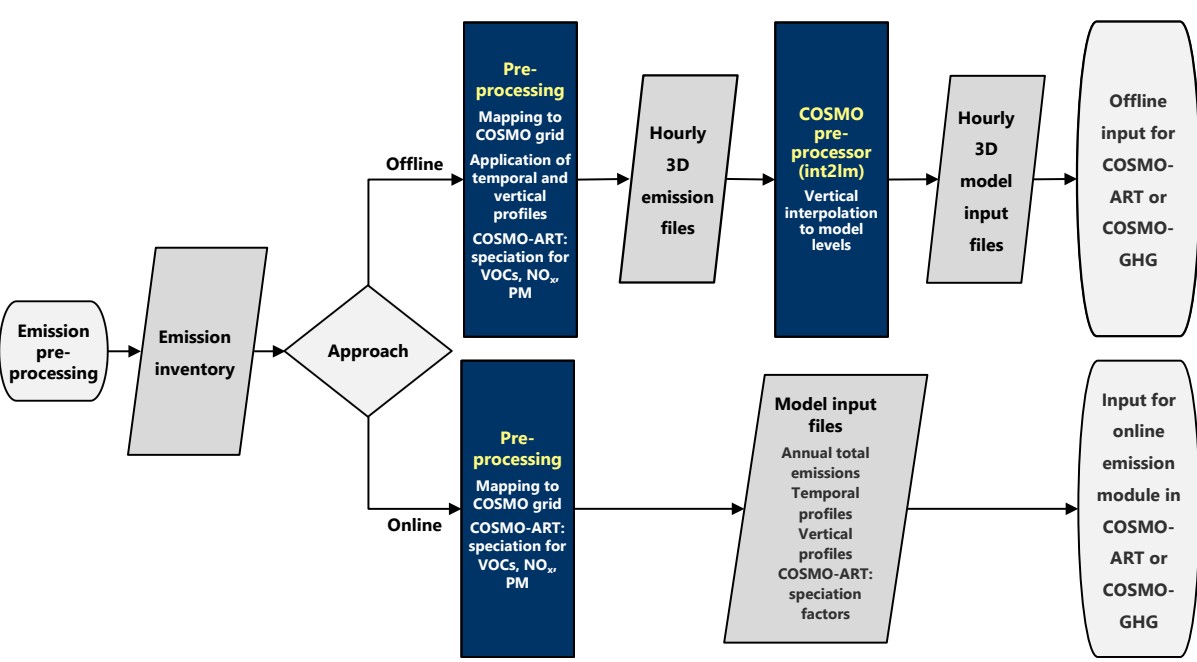

**Figure 1.** Flow chart of the emission pre-processing for both offline and online approaches as implemented in COSMO-GHG/ART. Dark grey parallelograms represent data (input or output). Blue rectangles are processing scripts. Start and end points are represented by ovals.

## 2 Emission pre-processing

For the pre-processing of the input data required by COSMO, a separate Python-based tool named "emiproc" has been developed. The tool provides the mapping of gridded emission inventories to any desired COSMO grid (latitude-longitude grid with a rotated pole) or regular latitude-longitude grid. Other projections could be implemented easily. It also generates the temporal

5 and vertical profiles needed for the online computation of emissions. If desired, the tool is able to generate offline emissions, which may be useful for users if no online emission module is implemented in their model.

The overall workflow of the emission pre-processing is illustrated in Fig. 1 for both approaches. In case of offline emissions, an additional step is required for the interpolation to the vertical grid of COSMO, which is executed by the official COSMO pre-processing tool int2lm http://www.cosmo-model.org/content/model/documentation/core/int2lm_2.05.pdf. Int2lm also merges

10 the emission fields with other COSMO inputs such as initial and boundary conditions and ensures proper formatting of the files.

### 2.1 Gridded emission inventories

Emission inventories of greenhouse gases and air pollutants are usually provided as annual 2D gridded fields per source category (e.g., traffic, industry, residential, agriculture) (Kuenen et al., 2014; Janssens-Maenhout et al., 2017; Crippa et al.,





2018). Before incorporating these emissions into a 3D atmospheric transport model, the 2D fields have to be convolved with temporal profiles of diurnal, day-of-week, and seasonal variability as well as with vertical emission profiles (Denier van der Gon et al., 2011; Pouliot et al., 2012; Kuenen et al., 2014). Current approaches for the definition of temporal and vertical profiles are often highly simplified (Matthias et al., 2018), partly due to a lack of detailed information but also due to the
limited flexibility offered by the offline approach.

Our current implementation of the module supports three different families of emission inventories: the inventories of the Netherlands Organization for applied scientific research TNO, the EDGAR inventories of the European Joint Research Centre (JRC), and the Swiss national emission inventories generated by Meteotest Inc., Switzerland. An extension to similar gridded inventories should be straightforward.

Although the inventories share a similar logic, there are distinct differences that had to be accounted for: In Europe, the two most prominent source classifications are the Standardized Nomenclature for Air Pollutants (SNAP; Centre on Emission Inventories and Projections (CEIP), 2018) and gridded Nomenclature for Reporting (GNFR; United Nations Economic Commission for Europe, 2015; Schindlbacher et al., 2016). SNAP is an old standard introduced in the context of the Convention on Long-Range Transboundary Air Pollution (CLRTAP), which had been used for many years in inventories produced by the European
Monitoring and Evaluation Programme (EMEP) and by TNO. The latest inventories, however, follow the EMEP/CORINAIR NFR, which was introduced to harmonize the source classification with the one used in National Inventory Reports to the United Nations Framework Convention for Climate Change (UNFCCC EEA, 2000). For gridded inventories, the large number of NFR categories are usually lumped into 13 individual classes (A–M). This reduced set of categories is commonly referred to as gridded NFR or GNFR. EDGAR inventories are reported as individual or lumped NFR categories, which are similar but
not identical to the GNFR standard. A table for unambiguous mapping between individual NFR, GNFR and SNAP categories is available at United Nations Economic Commission for Europe (2003).

Another difference is the map projection. EDGAR and TNO inventories are reported on a regular latitude-longitude grid, the Swiss Meteotest inventories on a Cartesian grid in the Swiss CH1903 Oblique Mercator projection. Furthermore, TNO inventories differentiate between point and area sources, which is not done in the other inventories. Point sources as reported to
the European Pollutant Release and Transfer Register E-PRTR (https://prtr.eea.europa.eu) are provided in the TNO inventories at their exact location, which has the advantage that in high resolution simulations the point sources can be assigned accurately to the respective model grid cell. In addition, this allows applying separate vertical profiles to point and area sources as shown in Sect. 3.3.2.

## 2.2   Temporal and vertical profiles

The emission of a tracer $X$ at time $t$ is calculated as the sum over the emissions per source category scaled with source-specific temporal scaling factors. A tracer is defined here as a quantity that is transported in the model, which can be a single trace gas or aerosol component, a composition of several species, or an idealized tracer. The temporal scaling factor for tracer $X$, source





category $s$ and time $t$ is generally given by

$$w_{X,s}(t) = w_{X,s,h}(h(t)) \cdot w_{X,s,d}(d(t)) \cdot w_{X,s,m}(m(t)) \tag{1}$$

where $w_{X,s,h}$, $w_{X,s,d}$ and $w_{X,s,m}$ are diurnal, day-of-week and seasonal scaling factors, respectively. The three scaling functions have a mean value of 1, such that the mean value of all scaling factors applied to a full year of data is one (or very close to one). The step functions $h(t)$, $d(t)$, and $m(t)$ are the hour of the day, the day of the week, and the month of the year corresponding to the continuous time $t$, respectively. The emission of $X$ at time $t$ is thus

$$E_X(t) = \sum_{s=0}^{n} E_{X,s} \cdot w_{X,s}(t) \tag{2}$$

where $E_{X,s}$ is the annual mean emission flux of $X$ of source category $s$ (which is the basic field usually provided by an inventory), and $n$ is the total number of source categories. This formula is applicable to an emission from a single grid cell or to a complete 2D emission field; i.e. $E_X$ and $E_{X,s}$ may be 2D fields. The functions may further depend on the country of the source. In that case, a further summation over countries is needed in combination with country masks. Our Python package and online emission module support country-specific time functions.

With this approach, real trace gases such as $CO_2$ can be simulated but also idealized tracers representing only a subset of sources, for example a tracer representing only traffic $CO_2$ emissions, by only summing over a subset of source categories in Eq. (2).

Emissions do not only occur at the surface but should be treated in 3D (Bieser et al., 2011b; Brunner et al., 2019). This is particularly true for elevated emissions from power plants or air traffic. Idealized vertical scaling functions $v_s$ are available for anthropogenic emissions, which distribute the emissions from a source of category $s$ over a discrete set of geometric vertical layers (altitude relative to ground). The scaling factors add up to 1 when summed over all vertical layers. Examples are given in the Supplement. The emission of the simulated tracer $X$ at time $t$ and in vertical layer $k$ is then given as

$$E_{X,k}(t) = \sum_{s=0}^{n} E_{X,s} \cdot w_{X,s}(t) \cdot v_{X,s,k}. \tag{3}$$

The vertical profiles do not depend on time $t$ in the current implementation of the module. This could be implemented in the future, for example to account for meteorology-dependent plume rise of emissions from power plants.

## 2.3 Speciation

The chemical compounds simulated in COSMO-ART include species for which inventories provide direct emission strengths (e.g. $SO_2$ or $NH_3$). However, for other species, the inventories only provide aggregated information for a family of compounds. This is the case for $NO_x$ (sum of NO and $NO_2$), NMVOCs, and particulate matter with a diameter of less than 2.5 $\mu$m ($PM_{2.5}$) and 10 $\mu$m ($PM_{10}$) (sum of various organic and inorganic aerosol compounds).

Therefore, to compute the emission of an individual compound simulated in the model, chemical speciation factors have to be applied to the total mass of the family. These speciation factors are specific for different source categories, since, for



example, the composition of NMVOCs, PM and $NO_x$ emissions is different for traffic and residential heating. Furthermore, the speciation factors depend on the specific chemical mechanism applied in the model, which determines the mapping between real and model-simulated species.

Starting from Eq. (3), the emission of species $X$ at vertical level $k$ additionally accounting for source speciation factor $f$ is
given by

$$E_{X,k}(t) = \sum_{s=0}^{n} E_{X,s} \cdot w_{X,s}(t) \cdot f_{X,s} \cdot v_{X,s,k}. \tag{4}$$

## 2.4 Mapping to COSMO grid

All emission data need to be mapped onto the simulation grid, which in case of COSMO is a rotated latitude-longitude grid. Such a mapping is not trivial, since simple interpolation is not mass-conservative, whereas conservative nearest neighbor
methods may lead to undesired stripes or other discontinuities. In order to avoid such issues and to accurately conserve mass, we determine for each grid cell of the inventory the cells of the COSMO grid intersecting it, compute the areas of intersection, and store the ratios between intersected and total area of the cell. Finally, for each model grid cell, the emission contributions of all computed intersections weighted by the corresponding ratios are summed up. A minor caveat is that the area of intersection is computed in degree[2] instead of true geometric areas. We found this to be an acceptable approximation with errors of less
than $0.1\%$ for inventory grid cells of dimension $0.1°$ x $0.1°$ at up to $60°$ latitude.

## 3  Implementation in COSMO-GHG and COSMO-ART

The online module was implemented in two extended versions of COSMO, COSMO-GHG developed for the simulation of passive tracers (e.g. greenhouse gases), and COSMO-ART for reactive trace gases and aerosols. An early version of the COSMO-GHG extension was developed in the CarboCount-CH project (Liu et al., 2017). It was built atop a generic tracer
module, which was introduced in COSMO version 5.0 to enable a flexible definition of tracers with specific properties defined by their metadata (Roches and Fuhrer, 2012). In the standard weather prediction version of COSMO it is used for the advective, convective and turbulent transport of all moisture tracers (water in the gas phase and in different hydrometeor phases). However, the life cycle of a tracer usually involves also other aspects not considered in COSMO such as emissions or removal from the atmosphere. In the GHG extension, emissions can be supplied in the form of 2D surface fluxes or 3D volume emissions. To
activate the GHG extension, COSMO has to be compiled with the `-DGHG` flag to enable the `#ifdef GHG` directives. Table 1 summarizes those interfaces to the GHG extension, including subroutine calls and number of `#ifdef` directives in the COSMO code.

Recently, the standard, CPU-based (Central Processing Unit) COSMO version released by the German Weather Service has been fully ported to GPUs (Graphics Processing Unit) (Fuhrer et al., 2014). This efficient GPU-enabled code, called COSMO-
POMPA (Performance On Massively Parallel Architectures), is used operationally by the Swiss Federal Office of Meteorology and Climatology (MeteoSwiss) for daily weather forecasting. COSMO-POMPA has been integrated into the latest official COSMO release 5.6a, which can be compiled for both CPU- und GPU-based systems.





**Table 1.** Changes due to the GHG extension in the COSMO code

| File name | Description | No. of ifdefs | Remarks |
|---|---|---|---|
| m_online_emissions.f90 | Contains subroutines for the reading in and computation of the gridded emissions vertical and temporal profiles for the online anthropogenic emissions module. | - | New file |
| acc_global_data.f90 | Performs allocation/deallocation of global fields on the accelerator using OpenACC directives. | 2 | 1 subroutine call |
| data_io.f90 | Contains all data necessary for input and output of grib or netCDF files. | 1 | 2 additional variables |
| lmorg.f90 | Main program. | 6 | 8 subroutine calls |
| organize_data.f90 | Organizes the I/O of the model. | 5 | 7 subroutine calls |

In order to benefit from the high efficiency of the GPU-enabled code, the GHG extension was ported to GPUs in the framework of the project SMARTCARB (Brunner et al., 2019). The porting was done using OpenACC compiler directives, which is a high-level approach to offload compute-intensive parts to a GPU accelerator (Lapillonne and Fuhrer, 2014). The same approach was employed for the porting of the online emission module. Since COSMO 5.6a can be compiled for both

CPU- and GPU-based platforms, the GHG-extension and the online emission module had to be programmed in a way that they can be executed on both platforms as well. Depending on the chosen platform, code sections related to OpenACC directives are included or excluded from compilation based on `#ifdef _OPENACC` compiler directives.

COSMO-ART was developed at the Karlsruhe Institute of Technology for the simulation of air pollutants and their interactions with meteorology (Vogel et al., 2009; Knote et al., 2011). The ART extension makes use of the same generic tracer

mechanism implemented in COSMO, but adds an additional layer (a structure `art_species`) allowing for a more comprehensive definition of tracer properties such as molecular weight, initial and boundary values, deposition properties and so on. For each ART species, a corresponding tracer is generated and dynamically allocated in memory. The position of the associated tracer is referenced by an index in the `art_species` structure. Because the ART extension has not yet been ported to GPUs, COSMO-ART does not run with COSMO 5.6a but with an older release 5.1.

**3.1 Basic framework for online emission processing in COSMO**

The main philosophy is to read in all input data required for the online emission module only once at the start of the simulation. These data include annual mean sector-specific 2D emission fields $E_{X,s}$ as well as the temporal, vertical and speciation profiles. During the simulation, these profiles are applied online to update the hourly emissions for each species according to equations 1-4.





**Table 2.** Members of the `GHGCTL` and `OAECTL` namelist group required in `INPUT_GHG` for COSMO-GHG and `INPUT_OAE` for COSMO-ART, respectively, for setting up the online emission module.

| Name | Description | Variable type |
|---|---|---|
| `in_tracers` | Number of `TRACER` groups (see Table 3) | INT |
| `vertical_profile_nc` | Filename for vertical profiles | CHAR |
| `hour_of_day_nc` | Filename for "hour of day" time profiles | CHAR |
| `day_of_week_nc` | Filename for "day of week" time profiles | CHAR |
| `month_of_year_nc` | Filename for "month of year" time profiles | CHAR |
| `hour_of_year_nc` | Filename for "hour of year" time profiles | CHAR |
| `gridded_emissions_nc` | Filename for gridded emissions | CHAR |
| `iemiss_interp` | Type of temporal interpolation (only COSMO-GHG) | INT |
| | `0`: constant emissions read for present full hour | |
| | `1`: linear interpolation of emissions between present and next full hour | |

**Table 3.** Namelist members of the `TRACER` group in `INPUT_GHG` and `INPUT_OAE`, respectively. Some namelist members are only implemented in COSMO-GHG or COSMO-ART.

| Name | Description | Variable type | Default value |
|---|---|---|---|
| `yshort_name` | Name of tracer | CHAR | `'undefined'` |
| `itype_emiss` | Type of 3D volume emissions (only COSMO-GHG): | INT | −999 |
| | `0`: no emissions | | |
| | `1`: offline emissions (i.e., read from file) | | |
| | `2`: online emissions | | |
| `itype_tscale` | Type of temporal scaling (only COSMO-GHG) | INT | 0 |
| | `0`: no temporal scaling | | |
| | `1`: temporal scaling using hour of day, day of week and month of year | | |
| | `2`: temporal scaling using hour of year | | |
| `ycatl(:)` | List of categories $s$ considered as sources of the tracer | CHAR | - |
| `ytpl(:)` | List of temporal profiles used for each element of `ycatl` | CHAR | - |
| `yvpl(:)` | List of vertical profiles used for each element of `ycatl` | CHAR | - |
| `contribl(:)` | List of speciation factors per category (only COSMO-ART) | REAL | −10.0 |

In order to implement new tracers flexibly, `TRACER` groups were added to the `INPUT_GHG` namelist file. A subset of the possible entries (members) of the `TRACER` group is presented in Table 3 and an example with a complete list of entries is given in the Supplement. For the online emission module, the definition of the `TRACER` group was extended with the following parameters:





1. A new possible value for the switch `itype_emiss`, which needs to be set to 2 to activate online emissions for this tracer.

2. A list tag `ycatl` listing the categories $s$ considered as sources of the tracer.

3. A list tag `ytpl` listing the temporal profile used for each element of `ycatl`.

4. A list tag `yvpl` listing the vertical profile used for each element of `ycatl`.

5. A list tag `contribl` used for chemical speciation, which lists the contribution of categories $s$ to the total emitted mass of the tracer (only COSMO-ART).

Note that the comma-separated lists `ycatl`, `ytpl`, `yvpl` and `contribl` need to have the same length. In contrast to COSMO-GHG, the definition of tracers is fixed (i.e. hard-coded) in COSMO-ART and, therefore, no namelist file `INPUT_GHG`

exists. In order to enable the same functionality as in COSMO-GHG, a new namelist file `INPUT_OAE` was introduced, where for each emitted tracer a corresponding `TRACER` group has to be defined (OAE = Online Anthropogenic Emissions). In order to enable or disable the use of online emissions in COSMO-ART, a new boolean switch `lemiss_online` was implemented in the namelist file `INPUT_ART`, which needs to be set to `.TRUE.` to activate the online emission module.

At the start of a simulation, the online emission module reads in the emission fields and the temporal and vertical profiles

from the following netCDF files generated by the Python tool (full paths of the files have to be specified in the namelist file `INPUT_GHG` or `INPUT_OAE`, see Table 2):

1. `gridded_emissions_nc`: 2D gridded fields $E_{X,s}$ of tracers $X$ for all source categories $s$ contributing to emissions of $X$ directly on the COSMO grid. The netCDF variable names need to be identical to the category names $s$ listed in `ycatl`. This file also contains a corresponding 2D country mask, which is an integer field with each grid cell being

assigned the number of the country that has the largest fractional area.

2. `hour_of_day_nc`, `day_of_week_nc`, `month_of_year_nc`: Time functions of diurnal, day-of-week and seasonal variations per tracer and source category. These scalings are provided in three separate files by default. However, it is also possible to provide only one file `hour_of_year_nc` with hour-of-year scaling factors by setting `itype_tscale` to 2. The netCDF variable names must be identical to those listed in the `ytpl` list for each cate-

gory $s$.

3. `vertical_profile_nc`: Vertical profiles per tracer and source category. The number of levels and their heights above surface can be set independently of the vertical structure of the COSMO grid. The netCDF variable names need to be identical to those listed in the `yvpl` list for each category $s$.

Temporal profiles are arrays with the two dimensions `time` (e.g. 24 different `hourofday` in case of a diurnal profile) and

`country`. If no country-specific information is available or desired, a uniform country mask with a single value for the whole



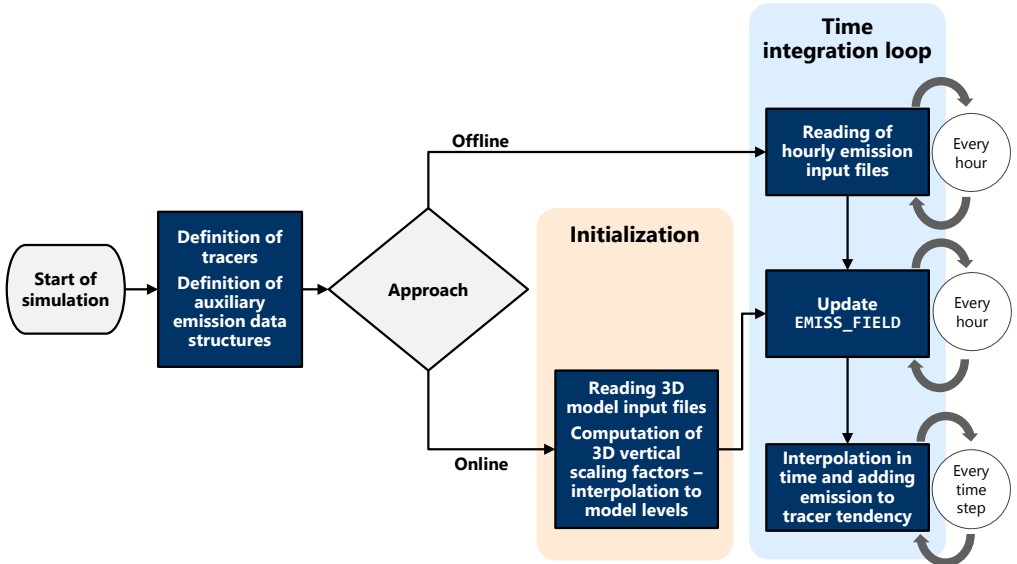

**Figure 2.** Flow chart for COSMO-GHG run-time processes both for the offline and online approach.

model domain needs to be generated. Vertical profiles, in contrast, are 1D arrays with `level` as the only dimension, since they are not expected to vary with country.

Speciation profiles for COSMO-ART are not provided in a separate netCDF file but are included in the namelist file `INPUT_OAE` through the list tag `contribl` of the `TRACER` group.

## 3.2 Modifications to the COSMO-GHG and COSMO-ART codes

In order to enable the computation of online emissions as an alternative to the default offline reading of emission files, the greenhouse gas module of COSMO-GHG (`m_online_emissions.f90`) was extended with new structures to store the information on temporal and vertical profiles and the annual mean sectorial emission grids. The arrays for temporal profiles and the emission grids are dynamically allocated as they depend on the tracer type. The information from the netCDF files is read at the beginning of the simulation. The vertical scaling factors, which are defined for layers above ground of fixed vertical extent (see Supplement), need to be translated into scaling factors for the vertical layers of COSMO. Since COSMO uses a spatially fixed grid with geometric hybrid vertical layers with thicknesses varying with the underlying topography, the vertical scaling factors are translated into dynamically allocated 3D arrays.

Before the implementation of the online emission module in COSMO-GHG, emissions were read in every hour from a file and assigned to an emission field associated with each tracer. In COSMO-GHG, fields that are conceptually attached to a prognostic variable of the model (e.g. an emission field attached to a tracer) are called associated fields. The definition, memory management and I/O aspects of associated fields are handled in `src_associated_fields.f90`. There, besides





the already existing functionality of reading in files, a mechanism was added to compute the emissions based on the dynamically allocated arrays for the emission grid and temporal and vertical scaling factors. Afterwards, no matter if online or offline emissions are used, the emission fields are updated at the beginning of each hourly interval. The updates account for the new temporal scaling for the current hour and the 3D vertical scaling factors calculated at the beginning of the simulation by

applying Eq. (3). The emissions are then kept constant during the current hour or linearly interpolated in time between the current and the next hour (if `iemiss_interp` is set to 1). The latter requires the computation (or reading) of the emission field at the following full hour. Finally, the emission field is added to the tracer tendencies at each model time step. All relevant computations in the GHG module use OpenACC directives in a similar manner as in other parts of the main COSMO code. A schematic of all relevant processes for offline and online emissions in COSMO-GHG during model run-time is shown in

Fig. 2.

Even though the concepts are very similar, the realization of the online emission module in COSMO-ART differed significantly from the implementation in COSMO-GHG. The emission module in ART (`art_emiss_prescribed.f90`) comes on top of the tracer module in COSMO and interacts closely with the tracer fields in other ART modules such as `art_mademod.f90` for aerosol processes. It was therefore desirable to keep most of the functionality of the standard emis-

sion modules to ensure the usability of both online and offline emission versions. The online emission module in COSMO-ART works as follows, focusing on the differences to the COSMO-GHG implementation: During initialization of the model, if the namelist parameter `lemiss_online` is set to `.TRUE.`, the required files (emissions, temporal and vertical scaling factors) are read in as described in Sect. 3.1. For every tracer with online emissions, the gridded emissions, the scaling factors (temporal and vertical), as well as speciation values, are collected in a data structure that associates this information with the location of

that tracer in the `art_species` structure. Then, at hourly intervals during the run, the emissions are calculated and written into the corresponding array in `art_species`. Because this replaces the assignment of emissions read in from external files to the same array as done in the standard offline version, no further changes to other modules are required. It is even possible to mix on- and offline emissions in a simulation run, such that emissions of certain tracers are read in, while others are calculated.

### 3.3 Practical examples

### 3.3.1 Merging of inventories

For regional scale simulations it is often desired to merge different inventories covering different regions, e.g. nesting the high-resolution Swiss national inventory into the coarser European inventory of TNO. Two different approaches have been implemented in the Python tool to tackle this use-case. For both approaches, the different inventories have to be mapped individually to the model grid using the Python tool as a first step. Afterwards, the inventories may be merged using a country

mask by overwriting the data of the coarser inventory with the data from the high-resolution one for all grid cells corresponding to the selected country (or region). A simpler approach is to make use of the fact that temporal profiles can be country-specific. To merge the two inventories during the online processing, the `hourofday` profile for the coarse inventory can be set to 0 for the country covered by the high-resolution inventory and vice versa for the high-resolution inventory.





### 3.3.2 Distinction between area and point source emissions

The inventories of TNO differentiate between area and point source emissions. Point source emissions as reported to the European Pollutant Release and Transfer Registry (E-PRTR) correspond to strong local sources, which are often emitted from tall stacks. An example is emissions from residential heating, which can originate from large centralized facilities (possibly
reported to E-PRTR) or from individual houses (reported as area sources). It may be desirable to apply different temporal and vertical profiles to these different types of sources. The Python tool allows generating separate grid maps and temporal and vertical profiles for point and area sources. Using these different maps and profiles for a given tracer is then straightforward, since point and area sources can be treated like separate source categories that need to be listed in the `ycatl`, `ytpl`, and `yvpl` tags of the tracer namelist group.

### 3.3.3 Modification of temporal and chemical speciation profiles

In a study of Athanasopoulou et al. (2017), the effect of the financial crisis in Greece on domestic fuel use and air pollution was investigated as an air quality modeling study using COSMO-ART. Due to the financial crisis, there was a sharp increase in wood burning for residential heating, which had major implications for air quality in Athens. Based on measurements of black carbon in the city, it became clear that the standard temporal profiles had to be updated to better reflect the actual burning
of wood. For example, the standard profiles for residential heating predicted peaks of similar amplitude in the morning and evening and a minimum during weekends. This largely contrasted with the measurements which suggested maximum usage during weekends and a major peak in the evening but not in the morning. Furthermore, the speciation of PM2.5 emissions had to be modified, since wood burning emissions contain a large proportion (approx. 80%) of primary organic aerosols. Adjusting the temporal and speciation profiles required changes to the emission pre-processing software and regenerating all hourly input
files for the simulation. Using the online emission module, the same could have been achieved by a simple replacement of the temporal profiles in the netCDF files and of the speciation profiles in the namelist file `INPUT_OAE`.

## 4 Performance evaluation (offline vs. online emissions)

Two sets of simulations comparing the results of online and offline emission handling using COSMO-GHG and COSMO-ART, respectively, are presented in the following. The comparison is made in terms of performance (computation time and storage
requirements) and in terms of simulated tracer fields. Ideally, the two approaches should lead to identical results, but as will be shown, small numerical errors e.g. due to truncation of floating point numbers were found to produce small differences.

### 4.1 COSMO-GHG

Using the COSMO-GHG model with online and offline setup, respectively, a simulation with a single $CO_2$ tracer for a domain covering the Alpine region at $0.01° \times 0.01°$ ($\approx 1.1$ km) horizontal resolution and with 60 vertical levels was conducted.
The simulation extended over a period of one week from 1 January 2019 to 8 January 2019. The boundary conditions for

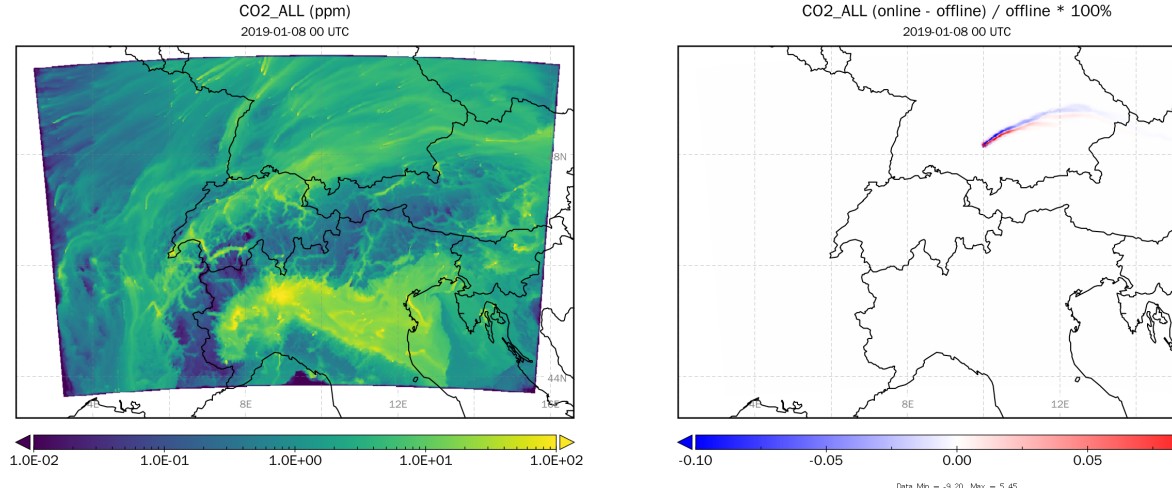

**Figure 3.** Surface-layer concentrations for the online simulation (left) and relative differences between online and offline simulations (right) of the $CO_2$ tracer corresponding to all anthropogenic emissions after seven days in the COSMO-GHG test case.

meteorological fields were taken from COSMO-7 reanalyses provided by MeteoSwiss. The $CO_2$ tracer field was initialized to 0.

The model setup closely followed the settings of the operational high-resolution COSMO-1 forecast of MeteoSwiss. The simulations were performed on 60 hybrid CPU-GPU nodes of the Piz Daint supercomputer of the Swiss National Supercomputing Centre (CSCS). Each node included an Intel Xeon E5-2690 v3 12-core CPU and an NVIDIA Tesla P100 GPU.

The yearly gridded anthropogenic emissions of $CO_2$ were taken from the TNO-GHGco inventory for 2015 available at $0.1° \times 0.05°$ resolution. They were split into 12 GNFR categories. The F category for road transport was further divided in 3 subcategories depending on the type of vehicles. Over Switzerland, the TNO inventory was replaced by a Swiss $CO_2$ inventory at $500\,\text{m} \times 500\,\text{m}$ resolution, which was created by the company MeteoTest in the framework of the CarboCount-CH project (Liu et al., 2017). Due to a different source categorization used in the Swiss inventory, those categories were mapped to the GNFR categories B, C, F, J and L. The temporal profiles applied to the emissions are described in the Supplement. No vertical profiles were applied in this example, but all emission were released at the surface. The namelist definition of the tracer in this simulation is described in the Supplement.

An example of the distribution of the simulated $CO_2$ tracer representing all anthropogenic emissions in the domain and of the differences between online and offline is shown in Fig. 3. The figure shows instantaneous near-surface $CO_2$ at the end of the 7-day simulation period.

The differences between online and offline are negligible (around $10^{-6}$ ppm) almost everywhere except for a plume in southern Germany. This difference can be explained by a point source in the TNO inventory, which is located exactly on the border of a COSMO grid cell. When generating online and offline emissions, two separate gridded maps of emissions need





to be produced. [1] Because of floating point truncation errors, this point source was attributed to two adjacent grid cells when processed with the online and offline approach. The spatial shift of this plume leads to differences of around $\pm 0.15$ ppm but the spatial mean remains almost constant (differs less than $10^{-7}$ ppm).

In terms of computation time the two COSMO-GHG runs were almost identical, as seen in Table 4. Disk usage, on the other hand, was dramatically reduced when using online emissions (about $3\%$ of the offline case). This benefit would have been even larger for a longer simulation period.

## 4.2 COSMO-ART

The online and offline emission approaches of COSMO-ART were compared by performing a test simulation over Europe with a horizontal resolution of $0.12° \times 0.12°$ ($\approx 13\,km$) and 60 vertical levels. The simulation was driven by meteorological fields from the European Centre for Medium-Range Weather Forecasts (ECMWF) Integrated Forecast System (IFS) model. The initial and boundary conditions for the chemical species were taken from the global MOZART-4 model (Emmons et al., 2010). A standard configuration with RADMK chemical mechanism, Volatility Basis Set (VBS) for organic aerosols and ISORROPIA-II scheme for inorganic aerosols was selected as described in Athanasopoulou et al. (2013). The configuration for the meteorology closely followed the setup of the operational European COSMO-7 forecasts of MeteoSwiss. The 24 h test simulation was started at 26 June 2015, 00 UTC and ended at 27 June 2015, 00 UTC. It was conducted on 16 CPU nodes of Piz Daint. Each node consisted of two Intel Xeon E5-2695 v4 processors providing a total of 36 cores per node. The simulation thus used a total of 576 cores.

The European anthropogenic emission inventory CAMS-REG-AP_v2_2 generated by TNO was combined with the Swiss national emission inventory for reactive gases and aerosols generated by Meteotest Inc. Both inventories were based on the GNFR source classification. The two inventories were merged using a country mask for Switzerland, i.e., following the first approach described in Section 3.3.1.

The COSMO-ART simulations with online and offline emissions have almost identical outputs for the same test simulation as shown in Fig. 4 for surface-layer concentrations of the gas phase species $SO_2$ and ethane and for particulate sulfate ($SO_4$, variable `VSO4J` in COSMO-ART). The differences for $SO_2$ and ethane are in the order of numerical noise. For $SO_4$ the differences are larger but still several orders of magnitude smaller than the absolute concentrations. Due to the complexity and non-linearity of aerosol chemistry within COSMO-ART, small differences may eventually build up in the course of the simulation. While maximum relative differences of $SO_4$ are up to 10% for a few grid points, the spatial mean of these differences are small with $0.044\%$. Further statistical values for $SO_4$ and other variables are provided in the Supplement.

Similar to COSMO-GHG, the computation time was quite comparable for the two versions, as seen in Table 4. The reduction in time spent on I/O in the online version is thus largely compensated by the increase in computation time. Disk usage for the online emission version was only about 1% of the usage for the offline version. Again, the benefit in disk usage would grow proportionally with increasing length of the simulation period.

---

[1]This is due to the fact that offline emissions are later pre-processed through int2lm which requires a slightly larger domain.





**Figure 4.** Surface-layer concentrations for the online and offline simulations, relative and absolute differences between online and offline simulations in the COSMO-ART test case for surface layer fields of (a) $SO_2$, (b) $SO_4$ and (c) ethane after 24 h.

## 5 Conclusions

A comprehensive online emission processing tool for atmospheric composition simulations was presented and demonstrated for two atmospheric transport models, COSMO-GHG and COSMO-ART. The tool is composed of a standalone Python package as well as a model-specific Fortran 90 module, which was integrated into the two models. The Python tool prepares the input

5   for the simulation (a small set of netCDF files), which consists of gridded emission maps per source category (projected to the model grid), and temporal and vertical emission profiles. The Fortran 90 module reads these inputs at the beginning of a simulation and updates the 3D emissions for each simulated tracer at regular (e.g. hourly) intervals taking into account the





**Table 4.** Simulation time and input data size for online and offline COSMO-GHG and COSMO-ART test cases.

|  | Online | Offline | Online / Offline |
| --- | --- | --- | --- |
| COSMO-GHG |  |  |  |
| Simulation time | 3 h 49 min | 3 h 57 min | 0.96 |
| Input size | 80 MB | 2800 MB | 0.03 |
| COSMO-ART |  |  |  |
| Simulation time | 19 min 34 s | 19 min 47 s | 0.99 |
| Input size | 59 MB | 5600 MB | 0.01 |

corresponding temporal and vertical emission profiles. For COSMO-ART, additional speciation profiles have to be provided to distribute the emissions of a family of compounds like NMVOC over the individual model species.

The Python package is independent of the specific model implementation and is also able to generate hourly emission fields for models using the traditional offline approach. The Fortran 90 module, in contrast, was specifically designed for the two
COSMO variants and will have to be adapted for other model systems.

The online approach greatly simplifies the setup of new model simulations, since a single set of a few input files is sufficient for all simulations on a given model grid irrespective of the simulated time period. In comparison to the offline approach, where a large number of emission input files has to be generated for a simulation, this reduces the storage requirements by orders of magnitude.

The performance of the online and offline approaches was compared in two test simulations conducted with COSMO-GHG and COSMO-ART, respectively. In both cases the total computation time was almost identical, suggesting that the additional time required for computing hourly emissions online within the model could be compensated by the reduced time spent on I/O. The time required for data pre-processing was not analyzed, but was clearly much lower for the online approach. Both approaches produced almost identical concentration fields demonstrating their equivalence, with tiny differences attributable
to floating point truncation errors.

The online emission approach was successfully introduced in all our COSMO-based model systems and, owing to its flexibility and reduced storage and data pre-processing requirements, greatly simplified our daily working procedures. The Python package is a valuable tool on its own, as it is independent of the specific model system, can be applied in combination with several popular emission inventories, and offers an accurate mass-conserving method for mapping emissions to a given model
grid.

*Code and data availability.* The Python package "emiproc" for emission pre-processing is publicly available through the C2SM GitHub organization (https://github.com/C2SM-RCM/cosmo-emission-processing). The repository also includes the vertical and temporal profiles used here. The emission inventories are not part of the repository. The EDGAR inventory is accessible from the European Joint Research





Centre (JRC). The TNO inventories can be accessed through the Copernicus product catalogue (https://atmosphere.copernicus.eu/catalogue). The Swiss national emission inventories are available from the corresponding author upon request. The online emission module has been implemented both in COSMO-GHG and COSMO-ART. Both Fortran modules are available on the C2SM organization on GitHub and can be obtained from the corresponding author upon request. Access to the COSMO code is restricted to COSMO licensees. A free license can

be obtained for research use following the procedure described at http://www.cosmo-model.org/content/consortium/licencing.htm.

*Author contributions.* MJ led the writing of the manuscript with contributions from all co-authors and generated the figures. MJ, DO, GK, JMH, and QM jointly developed the Python package. VC, GK, DO, JMH, MJ and KO jointly developed the Fortran modules. DB proposed the idea, surveyed the study and contributed to the writing. QM and MJ conducted the COSMO-ART simulations, JMH the COSMO-GHG simulations.

*Competing interests.* The authors declare that they have no conflict of interest.

*Acknowledgements.* We acknowledge the support from the following projects: SMARTCARB funded by the European Space Agency (ESA) under contract no. 4000119599/16/NL/FF/mg, CHE through European Union's Horizon 2020 programme under grant no. 776186, Carbosense4D funded by the Swiss Data Science Center, and Multi-scale air pollution modeling funded by the Swiss Federal Office for the Environment. The work was also supported by a grant from the Swiss National Supercomputing Centre (CSCS) under project ID s862.

Finally, we would like to acknowledge the contributions of Federal Office for Meteorology and Climatology MeteoSwiss, Swiss National Supercomputing Centre (CSCS), and ETH Zurich to the development of the GPU-accelerated version of COSMO.





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
