# Peer review of "An online emission module for atmospheric chemistry transport models: Implementation in COSMO-GHG v5.6a and COSMO-ART v5.1-3.1"

_Geoscientific Model Development, 2019_

## Referee Comment (RC1) · Anonymous Referee #1 · 10 Feb 2020

This manuscript presents a description of an online anthropogenic emission module and its implementation in two different atmospheric transport models, COSMO-GHG and COSMO-ART. The strength of the tool is (when compared to other existing emission processing tools) in its ability to perform online operations, which allow reducing the number and size of input files and the corresponding I/O. This is demonstrated by comparing the performance of the online emission module to an off-line and stand-alone Python package tool also developed by the authors. The paper is well written although it is sometimes lacking details. Therefore, it should be revised according to the following comments before its publication.

[Figure]

*Figure 1 and section 2:

Please indicate the file format of the data listed in the parallelograms.

If I understood it correctly, the python tool is responsible for generating the emission and temporal and vertical profiles NetCDF files that are later read by the online emission module. Nevertheless, in Figure 1 it is not clear which information/files are used to generate this NetCDF files (e.g. in which format are the emission and temporal/vertical profiles originally provided by the user to the Python tool). Similarly, the description and process for generating these NetCDF files (e.g. gridded_emissions_nc, hour_of_day_nc, day_of_week_nc, ...) should be included in section 2 and corresponding subsections (right now the description of the files is included in section 3.1).

*Section 2.1 – gridded emission inventories:

Perhaps this section should be accompanied with a table that list the different inventories that are currently available for processing and their main characteristics (e.g. name of inventory, pollutants considered, sector classification, year(s) of reference, spatial resolution-coverage, reference).

In its current version, the presented tool is capable of processing three different families of anthropogenic inventories (TNO, EDGAR and Swiss national inventory). I understand that other types of emissions such as biogenic or ocean are estimated/incorporated into COSMO using other specific models/tools (e.g. MEGAN for BNMVOCs), but what about biomass burning emissions (e.g GFAS or GFED)? Are they being estimated using an online approach inside COSMO?

*Section 2.2 and 2.3:

Temporal, vertical and speciation profiles are being applied to gridded emissions. Therefore, all the equations of these sections should represent emission of a tracer X at time t and grid cell c (instead of only tracer X at time t).

*Section 2.4:

Why the grid cell areas are calculated in degree? There are tools like CDO (Cdo{rbpy} for Python) that allow calculating grid cell areas in m2. It is currently not clear with which tools/libraries is the emission conservative mapping being performed (e.g using ESMF? Specific python libraries that allow creating geometric objects, such as shapefiles, and subsequently performing spatial interpolations?)

Following with the previous sections (2.2 and 2.3), an equation illustrating how the emissions are being mapped from the source grid cells onto destination grid cells should be included.

*Section 3.3.1:

How are separated the emissions when there are several countries into a cell (i.e. country border cells)? Or it is assumed that all emissions belong to the country that contains the largest fraction of the cell? Some inventories like TNO provide the information of emissions per grid cell and emitting country. Is this information used when given?

*Conclusion and Table 4 results:

According to the results presented in Table 4, the main advantage of using the online emission module instead of the offline tool is the significant reduction of the disk storage. Both in the COSMO-ART and COSMO-GHG test cases, authors suggest that increasing the length of the simulation period would imply a proportional growth of the benefit in disk usage. This very much would depend on how the workflow of the modelling system is set up. For instance, when running a 1-year simulation with an offline emission tool, modelers tend to generate an emission file of 24 hours for day "d", then run the atmospheric chemistry model for day "d", erase the emission file for day "d", generate an emission file of 24 hours for day "d+1", run the atmospheric chemistry model for day "d+1" and so on. In this situation, the benefit in disk usage would remain the same regardless of the simulation period (i.e. you will always be storing only one emission file of 24 hours). Having said that, I think the authors should include the time

required for data pre-processing when comparing the off-line and online approach. Otherwise the comparison remains unfair: the online computation of hourly emissions is considered but the offline computation of hourly emissions is not, which is also part of the modelling chain. Finally, the values reported for the simulation times should be split between time required to compute hourly emissions (online/offline), time spent on I/O and "others". This would allow to justify the suggested compensation between I/O decrease and computation increase reported by the authors.

*Meteorological parametrizations:

Besides the improvement shown by the authors in terms of disk storage, one would say that the main advantage of building an online emission module is the capability of computing on-line meteorological parametrizations such as heating degree day (for temporally distributing residential heating emissions) or plume rise calculation (for vertically distributing point source emissions). Scientifically (and even technically) speaking, the gains of performing such an implementation in comparison to an off-line approach may be much more significant. I understand that such an implementation may be more complex/time consuming than the one presented in this work, but perhaps it would be good to highlight it as a current limitation of the current tool and the need to work on it in the near future.

*Portability to other atmospheric transport models:

Authors suggest that the offline python package is of potential use for any atmospheric transport model system. I think this statement may be too strong in its current formulation, especially considering that each model requires the emissions to be provided following specific file formats (e.g. specific attributes in the NetCDF files) and conventions (e.g. units), and that certain models (e.g. WRF-CHEM, CMAQ) work with map projections that are currently not supported by the python package (e.g. lambert conformal conic, Mercator). Moreover, the current offline approach includes a COSMO pre-processor tool to perform a vertical interpolation to the model levels. How this

vertical interpolation would be performed in other models?

*Supplementary material:

Table S-1 to S-5: please specify from which works/references are the profiles derived.

COSMO-GHG example namelist includes some namelist members that are not defined in Table 3 of the manuscript or in the text (e.g. itype_lbc). Please provide a description of them (or remove them if they are not relevant to the publication).

In order to illustrate how "contribl" is used for chemical speciation, can you also provide a COSMO-ART example namelist?

*Other specific comments:

Please specify the units of $E_x(t)$, $E_{x,s}$ and $f_{x,s}$

Table 3: Why itype_emiss and itype_tscale are only needed for COSMO-GHG?

Figure 2 caption, please include the meaning of each polygon shape/colour (following the example of Figure 1's caption).

Please, add the following reference to the CAMS-REG-APV2_2 inventory: Granier, C., Darras, S., Denier van der Gon, H.A.C., Doubalova, J., Elguindi, N., Galle, B., Gauss, M., Guevara, M., Jalkanen, J.-P., Kuenen, J., Liousse, C., Quack, B., Simpson, D., and Sindelarova, K.: The Copernicus Atmosphere Monitoring Service global and regional emissions (April 2019 version), Copernicus Atmosphere Monitoring Service (CAMS) report, 2019, doi:10.24380/d0bn-kx16, 2019

In the introduction, when listing currently available emission processing systems, I would suggest to also mention the HEMCO system: Keller, C. A., Long, M. S., Yantosca, R. M., Da Silva, A. M., Pawson, S., and Jacob, D. J.: HEMCO v1.0: a versatile, ESMF-compliant component for calculating emissions in atmospheric models, Geosci. Model Dev., 7, 1409–1417, https://doi.org/10.5194/gmd-7-1409-2014, 2014

---

## Referee Comment (RC2) · Anonymous Referee #2 · 13 Feb 2020

The paper describes a new strategy for anthropogenic emissions management in chemistry-transport models. This strategy seems, in its principle, applicable to any CTM, but the processing tool proposed by the authors has been applied to two COSMO extensions (ART and GHG) but, as the authors acknowledge, would need additional work to be really usable with other CTMs. Their strategy essentially consists in splitting the emission processing in two steps, with horizontal redistribution performed offline, and vertical and time redistributions performed online, by the CTM itself.

The authors present essentially one python preprocessing module, emiproc, which performs essentially horizontal redistribution of the emissions from three types of invento-

ries onto COSMO (rotated) lat-lon grids, and the developments they have performed inside COSMO to enable online calculation of emissions. They convincingly show that the results of their model is not affected by this new approach.

Their manuscript is clear and very well written even though some points lack explanations. It contains no scientific novelty (nor does it pretend to do so), but in my opinion the methodology they propose and implement is clearly valuable and interesting for other teams and needs to be published.

I have four comments that in my opinion need to be adressed before publication.

1. The authors do not explain how they take into account the distinction between land and sea. No discussion is provided on that point. Lines 11-12, p. 6 suggest that the emission for any, possibly coarse, cell of the input inventory will be assumed to be evenly distributed within this cell, and thereafter evenly redistributed between all the intersecting model cells according to the intersection area. In the common case when model cell is smaller than inventory cell, this would lead to emissions of a clearly continental nature (e.g. traffic, residential heating) ending up in part in purely oceanic model cells, and shipping emissions ending up in part in purely continental cells. The authors should either explain how they avoid this problem, or acknowledge this point as a serious limitation of their tool that needs to be adressed in its future versions.

2. Similar to point 1 (but less hindering), accounting for landuse differences to have a meaningful distribution of emissions among model cells is also possible (e.g. the emiSURF emission preprocessing tool described in Mailler et al., 2017, "CHIMERE-2017 : from urban to hemispheric chemistry-transport modeling"). From a computational point of view this would be the same as accounting for land/ocean difference, but less critical. If the inventory is, e.g., 10x10km, one inventory cell could very well contain one area with forest and one city, and taking this information into account when generating emissions at the, say, 1x1km resolution of the model is clearly useful. Same as for point 1., if this is done the authors should explain how, and if not they should

acknowledge this point as a limitation compared to other tools.

3. I can take the word of the authors that it does not generate major errors, but the areas in square degrees are not really convincing. An "area" in squared degrees is not an area, and is not even proportional to the geometric area (that would depend on the orientation of the considered area, zonally or meridionally elongated). It would probably not be more difficult to work with real geometric areas on a sphere. The meaning of an area in squared degrees in really unclear to me, in particular if either the inventory or the model grid is not a regular lat-lon grid. What is a square degree close to the pole for example ? I feel unconfortable with this point which more or less prevents a clear and physical writing of the equations for horizontal redistribution of the emissions, which are the key points of the tool they propose. In my opinion, the code should be slightly modified to deal with real geometric areas in squared meters.

4. The country-dependance of time profiles is mentioned in the text but not in the equations. As a consequence, the management of cells that are at the border of two or more countries is not really made clear : does teh CTM have, for these cells, yearly amount of pollutant per snap sector *and per country* ? This problem is not major, but the authors should make clear how they deal with this issue.

---

## Author Comment (AC1) · 2 Apr 2020

**Reviewer 1**

**Reviewer Point P 1.1** — Figure 1 and section 2:

Please indicate the file format of the data listed in the parallelograms.

If I understood it correctly, the python tool is responsible for generating the emission and temporal and vertical
profiles NetCDF files that are later read by the online emission module. Nevertheless, in Figure 1 it is not clear which
information/files are used to generate this NetCDF files (e.g. in which format are the emission and temporal/vertical
profiles originally provided by the user to the Python tool). Similarly, the description and process for generating these
NetCDF files (e.g. `gridded_emissions_nc`, `hour_of_day_nc`, `day_of_week_nc`, ...) should be included in section 2
and corresponding subsections (right now the description of the files is included in section 3.1).

**Reply**: Figure 1 has been revised following the reviewer's suggestions. The file format of each dataset (grey parallelograms)
is now indicated. The input files for temporal and vertical profiles as well as speciation factors (for COSMO-ART) now also
appear in the first input data box. Data produced by the Python tool or int2lm are labeled with "output". Furthermore,
the overall formatting of the figure has been improved for better readability.

The description of the netCDF files generated by our Python tool has been moved to Section 2:

The generated netCDF files contain time functions of diurnal, day-of-week and seasonal variations per
tracer and source category. These scalings are provided in three separate files by default. However, it
is also possible to provide only one file with hour-of-year scaling factors. The temporal profile variables
are arrays with the two dimensions `time` (e.g. 24 different `hourofday` in case of a diurnal profile) and
`country`. If no country-specific information is available or desired, a uniform country mask with a single
value for the whole model domain needs to be generated. Vertical profiles, in contrast, are 1D arrays
with `level` as the only dimension, since they are not expected to vary with country.

Further:

The file generated by the Python tool contains vertical profiles per tracer and source category. The
number of levels and their heights above surface can be set independently of the vertical structure of the
COSMO grid.

In Section 3, a table has been added to link the variable names within these files with the namelist tags:

**Table 1.** Overview of netCDF files and corresponding namelist tags in the `TRACER` group in `INPUT_GHG` or `INPUT_OAE`. The
netCDF variable names must be identical to those listed in the namelist members (right column).

| Variable for netCDF file name | Namelist member |
| --- | --- |
| `gridded_emissions_nc` | `ycatl` |
| `hour_of_day_nc` | `ytpl` |
| `day_of_week_nc` | `ytpl` |
| `month_of_year_nc` | `ytpl` |
| `hour_of_year_nc` | `ytpl` |
| `vertical_profile_nc` | `yvpl` |

**Reviewer Point P 1.2** — Section 2.1 – gridded emission inventories:

Perhaps this section should be accompanied with a table that list the different inventories that are currently available for processing and their main characteristics (e.g. name of inventory, pollutants considered, sector classification, year(s) of reference, spatial resolution-coverage, reference).

In its current version, the presented tool is capable of processing three different families of anthropogenic inventories
5 (TNO, EDGAR and Swiss national inventory). I understand that other types of emissions such as biogenic or ocean are estimated/incorporated into COSMO using other specific models/tools (e.g. MEGAN for BVOCs), but what about biomass burning emissions (e.g GFAS or GFED)? Are they being estimated using an online approach inside COSMO?

**Reply**: From our point of view, an overview table of the different inventories would not be helpful, since we would like to
10 emphasize the flexible and generic nature of our implementation rather than presenting a tool that is tailored to specific types of inventories. The inventories currently supported are explained and referenced in the text. Furthermore, emission inventories change quite rapidly: new versions are released, formats change, additional years and species are covered, etc. The modularity and flexibility of our tool allows such changes to be incorporated rather quickly.

Certain types of natural emissions are indeed implemented in specific modules in COSMO-ART, but emissions from
15 natural fires are not. They have been accounted for in previous simulations using the traditional offline approach. Since biomass burning emissions do not follow a regular spatio-temporal pattern, our online emissions processing tool is not suitable for this type of emissions. We are aware that this is an important shortcoming. An extension of the tool to support a combination of offline inputs and online computation would be straightforward (since both methods are already implemented as separate options), but this is currently not implemented. We added the following sentence in the conclusions
20 section:

> Our tool is tailored to the processing of anthropogenic emissions, which follow a regular pattern in space and time, but it is not suitable for the processing of highly variable emissions such as emissions from biomass burning. This will require an extension to support a combination of offline inputs and online computation.

25 **Reviewer Point P 1.3** — Section 2.2 and 2.3:
Temporal, vertical and speciation profiles are being applied to gridded emissions. Therefore, all the equations of these sections should represent emission of a tracer X at time t and grid cell c (instead of only tracer X at time t).

**Reply**: In Section 2.3 (former Section 2.2) we state that "This formula is applicable to an emission from a single grid cell or to a complete 2D emission field; [...]" and thus, the additional index is not necessary here.

30 **Reviewer Point P 1.4** — Section 2.4:
Why the grid cell areas are calculated in degree? There are tools like CDO (Cdorbpy for Python) that allow calculating grid cell areas in m2. It is currently not clear with which tools/libraries is the emission conservative mapping being performed (e.g using ESMF? Specific python libraries that allow creating geometric objects, such as shapefiles, and subsequently performing spatial interpolations?)
35 Following with the previous sections (2.2 and 2.3), an equation illustrating how the emissions are being mapped from the source grid cells onto destination grid cells should be included.

**Reply**: To map emissions from the inventory to the model grid, we use our own implementation that makes use of the shapely and cartopy library. For each cell from the inventory and the COSMO grid, we produce a shapely polygon and use the "intersection" method to compute the intersection between the two. The intersection has an "area" method
40 returning the area of the intersection. Grid cell areas were only calculated in degrees for computing the overlap between a COSMO grid cell and an inventory grid cell. However, we modified the implementation to first project the polygons into the Mollweide equal-area projection, solving this issue. This has virtually not changed the results (maximum ratio difference of $3 \cdot 10^{-5}$).

We have moved this Section to 2.2 and revised it to clarify these points:

All emission data need to be mapped onto the simulation grid, which in case of COSMO is a rotated latitude-longitude grid. Such mapping is straightforward for point sources for which the emissions are added to the COSMO grid cells that contain the sources. For area sources this is less trivial, since simple interpolation is not mass-conservative, whereas conservative nearest neighbor methods may lead to undesired stripes or other discontinuities. In order to avoid such issues and to accurately conserve mass, we determine for each COSMO grid cell the relative fraction of the overlap with each inventory grid cell. The emissions for all source categories $s$ at each COSMO grid cell $i$ are then computed by

$$E_{i,s}^{\text{COSMO}} = \sum_{j=1}^{N} f_{i,j} \cdot E_{j,s}^{\text{inventory}} \tag{1}$$

where $E_{i,s}^{\text{COSMO}}$ and $E_{j,s}^{\text{inventory}}$ are the emissions (in mass/cell) at the $i$-th and $j$-th COSMO and inventory grid cell, respectively, $N$ is the number of grid cells in the inventory and $f_{i,j}$ is the dimensionless fraction of the source cell $j$ contributing to destination cell $i$. The fraction is determined by computing the intersection between each COSMO and inventory grid cell divided by the total area of the inventory cell using the equal-area Mollweide projection to conserve the mass.

Our tool does not redistribute the low-resolution inventory data onto the high-resolution COSMO-grid using additional information, such as land-sea masks and country boundaries, which can be used for improving the spatial allocation of area sources. This feature could be implemented in a future version similar to the implementation in CHIMERE-2017 model (Mailler et al., 2017).

The generated netCDF file contains 2D gridded fields $E_{i,s}$ directly on the COSMO grid. This file also contains a corresponding 2D country mask, which is an integer field with each grid cell being assigned the number of the country that has the largest fractional area. The unit of the emissions $E_{i,s}^{\text{COSMO}}$ depends on the actual model and are converted accordingly. For example, COSMO-GHG expects $\text{kg}\,\text{m}^{-2}\,\text{s}^{-1}$, whereas emissions in COSMO-ART are in $\text{kg}\,\text{h}^{-1}\,\text{cell}^{-1}$.

**Reviewer Point P 1.5** — Section 3.3.1:

How are separated the emissions when there are several countries into a cell (i.e. country border cells)? Or it is assumed that all emissions belong to the country that contains the largest fraction of the cell? Some inventories like TNO provide the information of emissions per grid cell and emitting country. Is this information used when given?

**Reply**: When running the COSMO online emission module, the assumption is made that the country mask attributes a single country to each grid cell. This country is indeed the one containing the largest fraction of the cell. However, this only impacts the temporal profile applied to the cell, which can be country dependent.

In regard to the emissions, if a grid cell from the TNO inventory covers multiple countries, there are indeed multiple entries in the inventory. Those are summed and the regridding is applied to the sum. Therefore, emissions from one country can be partially assigned to the neighbouring country and vice versa. This is also the case for cells bordering the sea, as noted in the Reviewer Point P 2.1. Note that point source emissions are still attributed to the correct COSMO grid cell. A mention of this limitation has been added to the Section 2.2:

Our tool does not redistribute the low-resolution inventory data onto the high-resolution COSMO-grid using additional information, such as land-sea masks and country boundaries, which can be used for improving the spatial allocation of area sources. This feature could be implemented in a future version similar to the implementation in CHIMERE-2017 model (Mailler et al., 2017).

**Reviewer Point P 1.6** — Conclusion and Table 4 results:

According to the results presented in Table 4, the main advantage of using the online emission module instead of the offline tool is the significant reduction of the disk storage. Both in the COSMO-ART and COSMO-GHG test

cases, authors suggest that increasing the length of the simulation period would imply a proportional growth of the benefit in disk usage. This very much would depend on how the workflow of the modelling system is set up. For instance, when running a 1-year simulation with an offline emission tool, modelers tend to generate an emission file of 24 hours for day "d", then run the atmospheric chemistry model for day "d", erase the emission file for day "d",

5    generate an emission file of 24 hours for day "d+1", run the atmospheric chemistry model for day "d+1" and so on. In this situation, the benefit in disk usage would remain the same regardless of the simulation period (i.e. you will always be storing only one emission file of 24 hours). Having said that, I think the authors should include the time required for data pre-processing when comparing the off-line and online approach. Otherwise the comparison remains unfair: the online computation of hourly emissions is considered but the offline computation of hourly emissions is

10    not, which is also part of the modelling chain. Finally, the values reported for the simulation times should be split between time required to compute hourly emissions (online/offline), time spent on I/O and "others". This would allow to justify the suggested compensation between I/O decrease and computation increase reported by the authors.

**Reply**: Thank you for this important comment. We agree that storage size can be controlled by a more sophisticated

15    workflow as you described. However, the amount of data that needs to be read in remains the same. This is why we refer to it as "input size" in Table 4 in particular. Reduced data storage is by far not the only advantage. From our view, the largest advantage is the greatly simplified workflow, since we don't need to produce new input files for every new simulation (unless we change domain size etc.). This is also what we state first when summarizing the advantages of the tool in the Conclusions.

20    To further specify the processing times required for pre-processing, we added another table showing the difference in all relevant pre-processing steps for online and offline processing, respectively. Unfortunately, the simulation times cannot be split since this information is not available.

**Table 2.** Benchmark for pre-processing online and offline emissions for the COSMO-GHG test case using the emiproc tool. Altogether, 19 categories from the emission inventories were processed (TNO: 14, Swiss: 5) as input for the $CO_2$ tracer. Pre-processing was performed on a local Linux cluster, using 14 threads in parallel. Processing times for generating the mapping and country mask file are excluded. Results are shown for different time periods.

| Processing step | Processing time (s) | | |
| --- | --- | --- | --- |
| | 1 day | 7 days | 365 days |
| Mapping and merging of inventories, profile generation | 78 | 78 | 78 |
| Offline Processing: extracting data from profiles (netCDF) | 106 | 106 | 106 |
| Offline Processing: generating offline files | 11 | 74 | 3859 |
| Total Online | 78 | 78 | 78 |
| Total Offline | 117 | 180 | 3965 |
| Ratio Online / Offline | 0.67 | 0.43 | 0.02 |

**Reviewer Point P 1.7** — Meteorological parametrizations:

25    Besides the improvement shown by the authors in terms of disk storage, one would say that the main advantage of building an online emission module is the capability of computing on-line meteorological parametrizations such as heating degree day (for temporally distributing residential heating emissions) or plume rise calculation (for vertically distributing point source emissions). Scientifically (and even technically) speaking, the gains of performing such an implementation in comparison to an off-line approach may be much more significant. I understand that such an

30    implementation may be more complex/time consuming than the one presented in this work, but perhaps it would be good to highlight it as a current limitation of the current tool and the need to work on it in the near future.

**Reply**: As pointed out in our reply to Reviewer Point P 1.6, the largest advantage of the tool is actually not the reduced storage demand but the simplified workflow. But we agree with the reviewer that a particularly attractive advantage of an online processing tool is the possibility to link the emissions to the actual meteorology in the model. This has indeed not yet been implemented and should be a focus of future developments. We added the following sentence at the end of the Conclusions section:

> Future developments will focus on the porting to other model systems such as ICON-ART and on the implementation of meteorology-dependent emissions such as emissions from residential heating or from lightning.

**Reviewer Point P 1.8** — Portability to other atmospheric transport models:
Authors suggest that the offline python package is of potential use for any atmospheric transport model system. I think this statement may be too strong in its current formulation, especially considering that each model requires the emissions to be provided following specific file formats (e.g. specific attributes in the NetCDF files) and conventions (e.g. units), and that certain models (e.g. WRF-CHEM, CMAQ) work with map projections that are currently not supported by the python package (e.g. lambert conformal conic, Mercator). Moreover, the current offline approach includes a COSMO pre-processor tool to perform a vertical interpolation to the model levels. How this vertical interpolation would be performed in other models?

**Reply**: We are convinced that adaptations to other model systems require only a small effort. If users want to generate emissions for different inventories and/or different transport models, they can implement their own classes within emiproc in a straightforward manner by modifying the existing examples. Also, any map projection available in the cartopy package can easily be utilized (including Lambert Conformal conic and Mercator). Units and attributes of the output netCDF variables are already model-dependent. For example, COSMO-GHG expects emissions in $\mathrm{kg\,m^{-2}\,s^{-1}}$, whereas in COSMO-ART it is $\mathrm{kg\,h^{-1}\,cell^{-1}}$, which is implemented in emiproc.

Note that the pre-processing does not interpolate the emissions to the vertical levels of the COSMO-model but to any pre-defined set of vertical levels. Interpolation to the terrain-following 3D grid of the COSMO-model is accomplished online as explained in Section 3.2. The same approach will have to be followed for any other model with terrain-following vertical coordinates.

Based on the reviewer's suggestion, we adjusted the text in the introduction referring to the offline tool as follows:

> Models and inventories that are currently not covered by the Python package can be implemented in a straightforward manner.

**Reviewer Point P 1.9** — Supplementary material:
Table S-1 to S-5: please specify from which works/references are the profiles derived.

COSMO-GHG example namelist includes some namelist members that are not defined in Table 3 of the manuscript or in the text (e.g. `itype_lbc`). Please provide a description of them (or remove them if they are not relevant to the publication).

In order to illustrate how "`contribl`" is used for chemical speciation, can you also provide a COSMO-ART example namelist?

**Reply**: The following sentences have been added to the supplement:

> Vertical profiles are based on source-specific profiles developed for the European Monitoring and Evaluation Program (EMEP) (Bieser et al., 2011) that have been modified for the SMARTCARB project (Brunner et al., 2019).

and

> The time profiles of emissions were originally published by TNO for SNAP emission categories (Denier van der Gon et al., 2011; Pouliot et al., 2012) and have been mapped to GNFR categories within the CHE project (Haussaire et al., 2018).

Originally, we wanted to provide a complete namelist example. However, as pointed out by the reviewer, some namelist members are not relevant for this publication. Thus, they have been removed from the example. Furthermore, we added a COSMO-ART example namelist for speciation of $NO_X$ into $NO_2$ to show the usage of `contribl`.

**Minor**

**Reviewer Point P 1.10** — Please specify the units of Ex(t), Ex,s and fx,s

**Reply**: Units for emissions, scaling and speciation factors have been added.

**Reviewer Point P 1.11** — Table 3: Why `itype_emiss` and `itype_tscale` are only needed for COSMO-GHG?

**Reply**: In COSMO-ART, the use of online (or offline) emissions cannot be specified for each tracer separately. Instead, there is a general Boolean switch `itype_emiss` to activate online emission processing for all tracers. COSMO-ART only supports method 1 of the namelist flag `itype_tscale`, i.e. "temporal scaling using hour of day, day of week and month of year". Therefore, the flag `itype_tscale` cannot be set. We added the following text in Section 3.1 to clarify these points:

> For this reason, the namelist switch `itype_emiss` does not exist in COSMO-ART. The same applies for `itype_tscale`, since temporal scaling is applied just for hour-of-day, day-of-week and month-of-year profiles.

**Reviewer Point P 1.12** — Figure 2 caption, please include the meaning of each polygon shape/colour (following the example of Figure 1's caption).

**Reply**: Done.

**Reviewer Point P 1.13** — Please, add the following reference to the CAMS-REG-APV2_2 inventory: Granier, C., Darras, S., Denier van der Gon, H.A.C., Doubalova, J., Elguindi, N., Galle, B., Gauss, M., Guevara, M., Jalkanen, J.-P., Kuenen, J., Liousse, C., Quack, B., Simpson, D., and Sindelarova, K.: The Copernicus Atmosphere Monitoring Service global and regional emissions (April 2019 version), Copernicus Atmosphere Monitoring Service (CAMS) report, 2019, doi:10.24380/d0bn-kx16, 2019

**Reply**: Thank you for this suggestion. We added this reference.

**Reviewer Point P 1.14** — In the introduction, when listing currently available emission processing systems, I would suggest to also mention the HEMCO system: Keller, C. A., Long, M. S., Yantosca, R. M., Da Silva, A. M., Pawson, S., and Jacob, D. J.: HEMCO v1.0: a versatile, ESMF-compliant component for calculating emissions in atmospheric models, Geosci. Model Dev., 7, 1409–1417, https://doi.org/10.5194/gmd-7-1409-2014, 2014

**Reply**: This is indeed an important reference that was missing. We added the reference as suggested.

**References**

Bieser, J., Aulinger, A., Matthias, V., Quante, M., and van Der Gon, H. D.: Vertical emission profiles for Europe based on plume rise calculations, Environmental Pollution, 159, 2935–2946, https://doi.org/10.1016/j.envpol.2011.04.030, http://www.sciencedirect.com/science/article/pii/S0269749111002387, 2011.

5   Brunner, D., Kuhlmann, G., Marshall, J., Clément, V., Fuhrer, O., Broquet, G., Löscher, A., and Meijer, Y.: Accounting for the vertical distribution of emissions in atmospheric $CO_2$ simulations, Atmospheric Chemistry and Physics, 19, 4541–4559, https://doi.org/10.5194/acp-19-4541-2019, https://www.atmos-chem-phys.net/19/4541/2019/, 2019.

Denier van der Gon, H., Hendriks, C., Kuenen, J., Segers, A., and Visschedijk, A.: Description of current temporal emission patterns and sensitivity of predicted AQ for temporal emission patterns: TNO Report, EU FP7 MACC deliverable report

10  D_D-EMIS_1.3, Report, MEP-R2003/166, Apeldoorn, The Netherlands, 2011.

Haussaire, J.-M., Brunner, D., Marshall, J., Prunet, P., Augusti-Panareda, A., Manders, A., Segers, A., Denier van der Gon, H., Maenhout, G., Houweling, S., and Krol, M.: CHE Deliverable 2.1 - Model systems and simulation configurations, Version 2.0, Tech. rep., $CO_2$ Human Emissions Project, https://www.che-project.eu/sites/default/files/2018-04/CHE-D2-1-V2-0.pdf, 2018.

15  Mailler, S., Menut, L., Khvorostyanov, D., Valari, M., Couvidat, F., Siour, G., Turquety, S., Briant, R., Tuccella, P., Bessagnet, B., Colette, A., Létinois, L., Markakis, K., and Meleux, F.: CHIMERE-2017: from urban to hemispheric chemistry-transport modeling, Geoscientific Model Development, 10, 2397–2423, https://doi.org/10.5194/gmd-10-2397-2017, https://www.geosci-model-dev.net/10/2397/2017/, 2017.

Pouliot, G., Pierce, T., van der Gon, H. D., Schaap, M., Moran, M., and Nopmongcol, U.: Comparing emission inventories and

20  model-ready emission datasets between Europe and North America for the AQMEII project, Atmospheric Environment, 53, 4 – 14, https://doi.org/https://doi.org/10.1016/j.atmosenv.2011.12.041, http://www.sciencedirect.com/science/article/pii/S1352231011013288, 2012.

---

## Author Comment (AC2) · 2 Apr 2020

**Reviewer 2**

Reviewer Point P 2.1 — The authors do not explain how they take into account the distinction between land and sea. No discussion is provided on that point. Lines 11-12, p. 6 suggest that the emission for any, possibly coarse, cell of the input inventory will be assumed to be evenly distributed within this cell, and thereafter evenly
redistributed between all the intersecting model cells according to the intersection area. In the common case when model cell is smaller than inventory cell, this would lead to emissions of a clearly continental nature (e.g. traffic, residential heating) ending up in part in purely oceanic model cells, and shipping emissions ending up in part in purely continental cells. The authors should either explain how they avoid this problem, or acknowledge this point as a serious limitation of their tool that needs to be adressed in its future versions.

10 **Reply**: Thank you for this important comment. Our current version does not use additional information (land/sea mask, landuse etc.) to redistribute low-resolution inventory data onto the high-resolution COSMO-grid. We have modified the section and now mention this limitation:

Our tool does not redistribute the low-resolution inventory data onto the high-resolution COSMO-grid using additional information, such as land-sea masks and country boundaries, which can be used for improving the spatial allocation of area sources. This feature could be implemented in a future version similar to the implementation in CHIMERE-2017 model (Mailler et al., 2017).

**Reviewer Point P 2.2** — Similar to point 1 (but less hindering), accounting for landuse differences to have a meaningful distribution of emissions among model cells is also possible (e.g. the emiSURF emission preprocessing tool described in Mailler et al., 2017, "CHIMERE-2017 : from urban to hemispheric chemistry-transport modeling").

- 20 From a computational point of view this would be the same as accounting for land/ocean difference, but less critical. If the inventory is, e.g., 10x10km, one inventory cell could very well contain one area with forest and one city, and taking this information into account when generating emissions at the, say, 1x1km resolution of the model is clearly useful. Same as for point 1., if this is done the authors should explain how, and if not they should acknowledge this point as a limitation compared to other tools.
- 25 **Reply**: See our previous response.

**Reviewer Point P 2.3** — I can take the word of the authors that it does not generate major errors, but the areas in square degrees are not really convincing. An "area" in squared degrees is not an area, and is not even proportional to the geometric area (that would depend on the orientation of the considered area, zonally or meridionally elongated). It would probably not be more difficult to work with real geometric areas on a sphere. The meaning of an area in squared degrees in north and the model with a not even proportional to area in squared degrees in north  $\alpha$  and  $\alpha$  area.

- 30 squared degrees in really unclear to me, in particular if either the inventory or the model grid is not a regular lat-lon grid. What is a square degree close to the pole for example ? I feel unconfortable with this point which more or less prevents a clear and physical writing of the equations for horizontal redistribution of the emissions, which are the key points of the tool they propose. In my opinion, the code should be slightly modified to deal with real geometric areas in squared meters.
- 35 **Reply**: We addressed this issue already in our reply to Reviewer Point P 1.4. We have modified the code to compute the intersection in an equal-area projection, solving this issue.

Reviewer Point P 2.4 — The country-dependance of time profiles is mentioned in the text but not in the equations. As a consequence, the management of cells that are at the border of two or more countries is not really

15

made clear : does teh CTM have, for these cells, yearly amount of pollutant per snap sector \*and per country\* ? This problem is not major, but the authors should make clear how they deal with this issue

**Reply**: We addressed this issue already in our reply to Reviewer Point P 1.5.

**References**

- Mailler, S., Menut, L., Khvorostyanov, D., Valari, M., Couvidat, F., Siour, G., Turquety, S., Briant, R., Tuccella, P., Bessagnet, B., Colette, A., Létinois, L., Markakis, K., and Meleux, F.: CHIMERE-2017: from urban to hemispheric chemistry-transport modeling, Geoscientific Model Development, 10, 2397–2423, https://doi.org/10.5194/gmd-10-2397-2017, https://www.geosci-model-dev.net/10/2397/2017/, 2017.
- 5